# Legal Aspects of Urban Water and Sanitation Regulatory Services: An Analysis of How the Spanish Experience Positively Would Contribute to the Brazilian New Regulation

**Asensio Navarro Ortega** [1,*] **and Rafael Burlani Neves** [2]

1   Department of Administrative Law, University of Granada, 18071 Granada, Spain
2   Public Law Department, University of Vale do Itajaí (Univali), 88136-170 Itajaí, Brazil; burlani@univali.br
*   Correspondence: asenavort@ugr.es

**Abstract:** This paper focuses on the legal and institutional framework of urban water services in Spain, emphasizing water sanitation by using proposals that would positively contribute to wastewater management in Brazil. The recent Brazilian Federal Law No. 14,026/20 aims to encourage investment in water sanitation, promoting public-private collaboration formulas so that service management is viable even in economically less-favored regions. In Spain, sanitation policies are aimed at fulfilling the set of obligations and objectives imposed by European Union Directives within the environmental policies of the Union. From an economic point of view, supply and sanitation water services are classified at European legal framework as "services of general economic interest" (SGEI), not subject to harmonized regulation and open to a natural monopoly provision regime, which they admit various types of management formulas, public and private, based on the ownership and public intervention of the service, both at national and European level. We believe that the Spanish experience in this field, beyond its singularities, can serve as a useful reference for Brazilian's urban wastewater new regulation for several reasons: (1) Because of the decentralized political scheme that both countries share and the need to articulate an adequate system of competencies in consequence; (2) Because of the international experience that Spanish companies have at the sector's technological forefront, they are very competitive; (3) Due to the adequate functioning of the Spanish legal and organizational framework since, despite its shortcomings, as we later will comment, it has managed to develop successful financing formulas and management models that, in general terms, have allowed to ensure with reasonable efficiency, continuity, stability and sustainability in the provision of urban water services.

**Keywords:** urban water management; sanitation; public policies; sustainable development; public contracts regulation; public-private collaboration

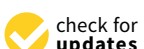

## 1. Introduction

This paper's purpose is to prospect structural perspectives of management and legal definition that contribute to universalization service, continuity and moderation (fair price) of wastewater treatment in Brazil. By facing a comparative analysis between public and private partnership policies in the European Union countries, especially in Spain. In the following pages, we deepen in what aspects Brazil would implement to improve its water quality policies in short term and provide the most efficient service to ensure its objectives beyond political interests and economic factors that underlie the wastewater treatment model's decision.

Water Sanitation is crucial for hygienic-sanitary management and environmental defense. Likewise, it is related to other economic and sectoral areas of the first magnitude: urban planning, tourism, industrial and commercial exploitation, authorization, or water quality. The United Nations (UN) estimates that 1 billion people lack sufficient water access, and 2.6 billion lack basic sanitation. Also, it triggers an alert by pointing out that by the

year 2025, a third of the world's population will not have access to drinking water to satisfy their basic needs. The 2030 Agenda for Sustainable Development foresees 17 Sustainable Development Goals (SDG) that must be reached by 2030. One of the objectives—number six—seeks to "ensure the availability and sustainable management of water and sanitation for all." [1]. The United Nations General Assembly proclaimed the 2018–2028 decade as the International Decade for Action, Water for Sustainable Development, which started on World Water Day, 22 March 2018, and ends on World Water Day, 22 March 2028. The UN resolution emphasizes that sustainable development and integrated water resources management are crucial to achieving social, economic and environmental goals. The document highlights the importance of implementing such programs and projects and promoting partnerships and involving various stakeholders to fulfil the 2030 Agenda for Sustainable Development, focusing on implementing SDG 6 of "ensuring the availability and sustainable water management and sanitation for all" [1]. Water scarcity—exacerbated by climate change and water-related disasters—can cause tensions that can become violent conflicts between people, communities and countries. Sustainable Development Goals (SDG) number 6 is also vital for preventing conflict and maintaining peace.

The Resolution of the United Nations General Assembly, 28 July 2010 (A/RES/64/292) recognizes access to water and sanitation as a fundamental human right for the full enjoyment of life and necessary complement for reaching other fundamental rights [2]. In its Resolution of 6 October 2010 (A/HCR/RES/15/9), the General Assembly encouraged the States to include appropriate instruments for access to this service in their regulations. However, the proliferation of armed conflicts, the rise of terrorism, uncontrolled population growth, massive migrations, climate change, famines caused by lack of water and food or the increased risk of contracting infectious and contagious diseases undermined the achievement of this Right, limiting its effectiveness. Still, to this day, more than 2.6 billion people do not have access to basic sanitation, so that every year around 1.5 million children under five years of age die and 443 million school days are missed due to diseases related to water and sanitation [1]. Overcrowding and survival in subhuman conditions are frequent in populations with no access to drinking water or basic sanitation. Barlow (2015) discloses the urgency of sanitation management for humanity, explaining how 70% of the population runs or risk to live without adequate sanitation in 2030 [3].

Fortunately, neither Brazil, as an emerging country, nor Spain, as a developed country, are in the precarious situation that we have just described. Each in its proportion, must undertake ambitious policies and investments both in terms of supply and, above all, sanitation.

## 2. Materials and Methods

The paper carries out a deductive logical analysis by studying cases and the legal literature and bibliographic references at a national and international level. In Brazil's case, materials from the National System of Information on Sanitation (SNIS, in Portuguese), have been consulted. Furthermore, in the case of Spain, the XVI National Study on the Supply of Drinking Water and Sanitation, elaborate by the Spanish Association of Water Supply and Sanitation (AEAS, in Spanish) and the Spanish Association of Urban Water Services Management Companies (AGA, in Spanish), which reaches 1795 municipalities and 33.88 million inhabitants, 73% of the Spanish population.

Pasqualetto et al. indicate that the comparison between water resources policies improves the planning and management mechanisms since the legal framework supports public policies in water resources [4].

The analysis focuses on the contribution of one system to other by taking elements from a legal system (Spain) that may contribute to another (that of Brazil). The results pretend to form a perspective to infer the management of basic sanitation considering the equal distribution of services, the power of contaminants and water pollutants, avoiding waste of water, and aspects for thinking in the implemented legal models.

The following assumptions justify the comparison between Brazil and Spain:

From a political and legal-administrative perspective, both countries have different hydrographic and climatic conditions sharing a series of common features. Thus, both are politically decentralized States and with important powers of the federated States and the Autonomous Communities. In both cases, water management is based on hydrological planning; and the public domain forms the basis for the legal consideration and regulation of waters. Likewise, water quality is increasingly important, and the hydrographic basin is considered a specific reference when constructing water management in both cases. To this purpose, we find antecedents in the legal literature that specifically compare water management in both countries [5].

In the field of urban water services, the municipal competence of the service, the defragmentation of regulation models and the search for economies of scale in both countries are common features in the comparison. The choice of the model is an eminently political discretionary decision but based on legal and technical foundations.

The example of Spain may be useful to introduce European policies on environmental matters and to approximate concepts of great relevance and meaning from the economic and legal optic (consideration of regulated urban water management services as economic services of general interest) within the framework of general liberalization, public contracting and defense of competition.

Spanish companies are global leaders in comprehensive water management and are pioneer at the forefront of water management technologies. Many of the main desalination infrastructures in the world are built and managed by Spanish companies. This is recognized by the Spanish Ministry of Foreign Affairs and Cooperation [6]. As it explains, seven of the twenty largest desalination companies are Spanish. A Spanish company manages the largest treatment plant in the world, among other examples of success. The Spanish water governance system is an international benchmark and an example of success in the Mediterranean region, being one of the world leaders in all management phases of the integral water cycle.

It cannot be said that the public or private formula is better in Brazil or Spain. The choice of model is related to the scale of security and service provision. The main strength of the Spanish regulation model has been not to think in terms of administrative borders, but to treat the space, the physical environment, where people live and work, adapting the model to the technical characteristics available to the municipality: altitude, orography, availability of the resource, governmental entities, number of inhabitants, infrastructures, etc. Although the decision is ultimately political, an attempt has been made to justify it based on technical and legal reasons and arguments, such as efficient service management or the search for economies of scale. The Spanish legislator only allows the direct management of water supply and sanitation when it is a more efficient option than indirect management and is sustainable and effective, applying economic profitability criteria. On this side, the Spanish model clearly opts for public and private collaboration models.

- Special attention must be paid to the necessary investment in the renovation of infrastructures or public assets in general.
- It is important to establish a correct coverage of costs with financing models that favor the system's sustainability.
- It is plausible to develop better specific legislation on the provision of urban water cycle services, not only at the sanitation level. In Brazil's case, it could be interesting to introduce a general regulator (specific for basic sanitation; with exclusive dedication) that, without modifying current competencies, controls and technically reviews the provision of the service, improving the levels of transparency.
- Innovation and public-private collaboration should be promoted, attracting the technical knowledge and know-how of private companies.

Under these parameters, we conclude the comparison by establishing a series of recommendations that can help in the implementation of the new Brazilian regulation on wastewater management and sanitation policies as a representative field of research

that requires comparative law solutions. In this sense, Brazil can benefit from other more popular European experiences, such as Spanish.

Authors keep an academic relationship studying this field. During the last five years various research actions were developed with the support of their partner institutions and scientific centers, either by participating in congresses, seminars, dissertations, both in Brazil and in Spain, or collaborating in publications related to the management of public policies and the regulation of water resources.

The authors are aware of the difficulty of comparing two very different regimens, which serve particular contexts, and the complexity of extrapolating results mechanically. On the contrary, we aspire to introduce, through critical analysis, a constructive vision based on the comparison between legal systems, to reflect on whether it would be possible to apply in Brazil some of the solutions that have been effective in Europe and, more specifically, in Spain.

## 3. Results

The research emphasizes in public and private partnership improving the public responsibility system, and effectively adapting the regulatory and institutional structure to manage the basic water sanitation service. The water sector integrates aspects related to the circular economy, investments, social action mechanisms, health, economic and environmental sustainability, etc. It is a strategic area that public authorities strive to improve.

The generally successful Spanish decentralized and atomized regulatory model may be useful for Brazil's experience. Among the most relevant issues, we highlight a series of factors:

- The international experience and the high technological development of Spanish companies, and the creation of formulas for public-private collaboration.
- The commitment to transparency and the establishment of accountability instruments as part of the service contracting strategy, following European regulations.
- The debate about the need to establish a more operational control of the concessionaires, independent of the direct or indirect management model.
- The convenience of integrating environmental regulations, public works and public procurement, and developing general and specific planning instruments in wastewater treatment.
- The importance of promoting competition within the limits of regulation of the service, as well as working on a geographical and non-administrative scale, as well as generating economies of scale that reduce the effective cost of the service.

On the other hand, the paper also aims to highlight two urban water management mistakes in Spain and their possible usefulness for Brazil institutions.

First, avoid political criteria interference to decide the urban water management model in each local government instead of attending economic and technical measures. An effort should be made to not determine the model according to ideological approaches by managing the information with transparency to prevent the decision from behaving like an arcane of statistical, accounting and budgetary data that do not transcend public opinion.

Second, facing the severe deficit of structural investment in infrastructure and redesign the financial and budgetary regime.

Finally, we analyze Brazil's suitability to create an independent regulatory authority that unifies the system and ensures economic and sustainability, as well as improves transparency, offers neutrality and legal security, monitoring the national sanitation strategy in Brazil.

The authors consider the convenience of thoroughly treating the related aspects of water supply and wastewater treatment since these services are provided or contracted in bulk, so their management is linked. In this way, this paper faces relevant factors that could positively contribute to establishing a more efficient regulatory model in Brazil, as follows (Table 1).

**Table 1.** Comparative factors that may contribute to a more efficient water and sanitation system development.

| Strengths of Spanish Legal Framework | Reason of Inefficiency of the Factors Cited in the Case of Brazil |
| --- | --- |
| Decentralized Political System | Brazilian new regulation centralizes sanitation strategies in the ANA agency that takes control of water policies for Brazil. |
| International & technical experience as a successful formula for public-private partnership. | In Brazil, basic sanitation service is mostly a state competence. More adaptative management could provide greater adequacy and private participation, making basic sanitation more efficient in Brazil. |
| Awareness of the need for control of the concessionaires. | Uncontrolled follow-up and monitoring of concessionaires (usually public companies) is notorious in Brazil. |
| Strengthening of anti-corruption mechanisms | There is no relevant information about the impact of corruption in this sector. The system's inefficiency and the cases of corruption in similar segments of activity would indicate a significant gap to reach basic sanitation in Brazil. |
| Transparency and accountability mechanisms. | Public institutions are making changes and doing things better, such as ANA and SNIS. In collaboration with civil society entities (Instituto Trat Brasil). However, a further improvement is necessary on this topic. |
| Ensures basic sanitation regardless of economic profitability—promotion of competitiveness. | In Brazil, the politic and governance greatly influence this issue. For now (2021), unprofitable basic sanitation operations are at risk of non-implementation. |
| Advanced legal framework and development of environmental law instruments. | Brazil's legal framework requires a holistic evolution in terms of efficiency and procedural security |
| Develop specific environmental planning to set tangible commitments and overcome the deficit of investments | Brazil may operate under financial instruments and specific planning to reduce the prospective structural deficit of investment. |
| Convenience of working based on technical monitorization of the service and the creation of economies of scale | Brazil goes deep on the concept of economy of scale and the effective cost of the service, bringing the management model's decision closer to more fruitful and profitable approaches. |
| Creation of an independent authority as an element of standardization and predictability of the system | Brazil should create an independent technical authority that moves the decision away from political parameters and introduces technical rationality and control. |

## 4. Discussion

*4.1. Water Policies in Brazil: Challenges and Complexities in the New Brazilian Federal Law No. 14,026/20*

According to Caubet [7], Brazil has a large water resource availability (something like 12% of all fresh water on the planet), including groundwaters. As a "continental country" Brazil owns enormous water reserves, having great supply capacity. However, these resources are not inexhaustible, and they cannot be enough if they are not well managed or protected from pollution. They are also not well distributed in the territory. Therefore, the price paid by the user for water supply and wastewater treatments is high. There are risks in the continuity of this service's provision (lack of legal guarantees). In this order, there is an evident need for public policies to preserve natural goods and provide an equal distribution of sanitation services. Likewise, the Brazilian Administration must improve the techniques for managing the pollutants in water and control the growing demand for the resource facing the increase in population, which also entails increasing diffuse pollution sources in urban areas or run-off from agricultural land.

Between all of them, the Guarani Aquifer is the most representative freshwater reservoir and reaches the territories of Argentina, Brazil, Paraguay and Uruguay, with an estimated extension of 1.2 million km$^2$, of which 840,000 km$^2$ are located in Brazilian territory, containing an accumulated volume of water of about 45,000 km$^3$.

Hussein [8] explains that Brazil has a hydro-hegemony in the South American region due to its economic, military and geopolitical influence. Another factor of political size over water, whether urban or not, is its water capacities to make money and investments.

This potential brings obligations and commitments, such as access to drinking water and basic sanitation.

According to D'Isep [9], water dignity is achieved by respecting quality (water may be drinkable); the quantity, that is, enough for survival; the priority of human access, in case of scarcity; and gratuity—at least about the minimum necessary for human survival. Considering the water ecosystem in Brazil, there are 12 hydrographic regions, impacted by Brazilian regulation.

The challenge to make public water management policy in Brazil involves some factors: the size of the country, with different biomes and with social inequality among the largest in the world [10]. Such aspects in themselves, dimension the greatness of the task, which requires a broad effort of sustainable public policy, which universal access is still an urgent measure—even more in times of pandemic. Along this path, there is a need for broad cooperation between the State, capital and society so that in the institutional spectrum, considering the actions of financing, management and social participation.

Da Silva et al. [11] also emphasize the importance of each nation's political objectives, according to the environmental policy in execution, since the intervention through, for example, infrastructure and constitution of norms also influences the destinies that each policy search as purpose.

One of the major problems is the non-treatment of wastewater, which dramatically affects both surface and underground sources (Table 2). Brazilian Federal Law No. 14,026/20, Art. 1, updates the legal framework for basic sanitation and amends Law No. 9984, of 17 July 2000, giving the National Water and Basic Sanitation Agency (ANA) the power to establish standards for the regulation of public sanitation.

In Brazil, 19 million people living in urban areas do not have access to drinking water. Another 21 million, living in rural areas, also do not have access to treated water, and only 46% of Brazilian households have sewage collection [12]. Some years ago, São Paulo's city faced hard rationing caused by the lack of rain and the pollution and contamination of the water bodies that served as supply due to the low water treatment. Table 2 below refers the situation of urban wastewater in Brazil:

**Table 2.** Annual water and sewage diagnosis, 2019 [13].

| BASIC WASTEWATER & SANITATION DATA IN BRAZIL | | | | Targets to Implement Measures |
|---|---|---|---|---|
| Diagnosis | 2011 | 2017 | 2019 | 2033 |
| Loss of water in the distribution | 38.8% | 33.3% | 39.2% | 31% |
| Population with treated water | 82.4% | 83.5% | 83.7% | 99% |
| Population with sewage collection | 48.1% | 52.4% | 54.1% | 90% |
| Investment in billions of euros | 1.67 | 1.67 | 2.009 | 112 (along the whole period) |

From a legal perspective, competences of wastewater and sanitation services was analyzed by the country's Supreme Federal Court (SFC) by ADI No. 1842-RJ, that supervise the Law that created the Metropolitan Region of Rio de Janeiro. The Brazilian Supreme Court stated that the municipality's ownership will be constituted when the basic sanitation cycle takes place entirely in its territory. The municipal entity can capture, supply water and fulfil sewage treatments to the domestic areas, considering the collection, transportation and operating. Otherwise, when the municipality cannot be responsible for the entire basic sanitation cycle, the holder will be a collegiate (in the form of a public consortium) representative of the metropolitan region or micro-region in which the municipalities are inserted.

In summary, in Brazil, the Federal Union will not be the holder of the basic sanitation service, municipalities or the metropolitan region will have such Right and duty. The new Brazilian basic sanitation law—Federal Law No. 14,026/20, in its Art. 7, by changing Art. 8 of Federal Law No. 11,445/07, recognized this interpretation determined before by the Brazilian Supreme Court.

Article 8 of the Federal Law provides that under the Federal Law, related to the provision of sanitation services, the State, together with the Municipalities that effectively share operational facilities that are part of metropolitan regions, urban agglomerations and micro-regions, will act together in the case of common interest. The exercise of ownership of the sanitation services may also be carried out by associated management, through a public consortium or cooperation agreement, under Art. 241 of the Federal Constitution, generating specifics ways of an "inter-municipal autarchy". These inter-municipal consortiums of basic sanitation will have as their sole objective of financing the initiatives for the implementation of structural measures for the supply of drinking water, sanitary sewage, urban cleaning, solid waste management, drainage and rainwater management, with no formalization of a program contract with a mixed capital company or public company, or the sub-delegation of the service provided by the inter-municipal authority without prior bidding procedure. The Sanitation Law pretends to reach a more financial sustainability and preferably include at least one metropolitan region, provided their integration by holders of sanitation services (§2). The holder of public sanitation services must define the entity responsible for regulating and inspecting these services, regardless of the type of provision (§5).

The Federal Union of Brazil can, through the country's constitutional competences, institute guidelines for basic sanitation, considering urban development: Art. 21 says the Union is responsible for "instituting guidelines for urban development, including housing, basic sanitation and urban transport". There is a space for the Union to act by issuing laws under basic sanitation in this context. However, this limit is very tenuous, as holders of this public service, basic sanitation can also legislate on the subject, which causes an evident overlap of sometimes conflicting rules. The issue is highly complex and involves distinguishing what, objectively, is being legislated. The normative context may be specified, whether in the technical question (types of technology; management and operation goals for basic sanitation; service price; quality indicators; others); or in the legal aspects (a form of contracting; the scope of service; quality of service; governance model; period of service; guarantees of service provision; legitimacy conditions for provision). Furthermore, pointing out guidelines and general guidelines.

On the other hand, some issues depend exclusively on the holders of essential sanitation services' actions. They are accurate Public Policy decisions, which imply optimizing critical sanitation management by the competent Public Administration.

Managing and defining the destination of financial investments does not mean that the holders of the basic sanitation service will be obliged to use these resources to implement their water and sewage management agendas. However, they may follow the guidelines instituted by the constituted authority to define reference standards, in this case, the National Agency of Water and Basic Sanitation (ANA) of Brazil (linked to the federal government). The fact is that ANA's resource fund will have a large volume of financing for basic sanitation in the country, making it very difficult for water and sewage service providers to seek another source of funds. It is a "binding pecuniary force", which in practice means "whoever does not follow the reference rules dictated by ANA will not have access to the Union's resources".

It is convenient to highlight that ANA had its attributions expanded (Article 2 of Federal Law No. 14,026/20). In addition to the mission of taking care of national water resources, it became responsible for establishing reference standards for public sanitation regulation.

This role in establishing reference standards falls on three matters: the technical, legal and political issue. In the legal field, ANA will regulate the standardization of business instruments to provide public sanitation services between the holder of the public service and the delegate entities.

In the technical area, it will regulate quality and efficiency standards in the provision, maintenance and operation of basic sanitation systems; the tariff regulation of public sanitation services; establish the criteria for regulatory accounting; the progressive reduction

and control of water loss; the reuse of treated sanitary effluents, by environmental and public health standards.

Third, the political perspective will define the goals of universalization of public essential sanitation services; define the methodology for calculating indemnities due to investments made and not yet amortized or depreciated; regulate regulatory bodies' governance.

Another important topic of the new Law is that the provision of public sanitation services, which is not part of the holder managed, will depend on prior bidding, with the subsequent conclusion of a concession contract, in which these must determine goals of (1) expansion of services; (2) reduction of losses in the distribution of treated water; (3) quality in the provision of services; (4) efficiency and rational use of water, energy and other natural resources and (5) reuse of waste.

The contracting of a third entity (indirect management) that is not part of the holder's Administration will be prohibited in the model of program contract, agreement, term of partnership or other instruments of a precarious nature. Those current regular program contracts remain in effect until the end of their contractual period. It should be noted that the new law prohibited the distribution of profits and dividends, of the contract in execution, by the service provider that is not complying with the goals and schedules established in the specific contract for the provision of public basic sanitation services.

Checking these challenges and complexities of the new legal framework for basic sanitation in Brazil, one wonders: can the experience of the European community, especially that of Spain, offer recommendations and perspectives allowing to assist the Brazilian experience in deciding. Whether the new legal definitions permeate the best way to provide the service, either directly, or indirectly by public or private operator. What is the more adequate partnership and public contracting model; What is the inspection model; What is the dispute resolution model; What is the tariff structuring model; Which model to choose (bidding), if based on price, service quality and environmental quality criteria; per sustainability requirement.

This study assumes that yes, the Spanish experience can contribute to the implementation of public policies on basic water and sanitation in Brazil, since the principles of universality, continuity, and more efficient water supply and wastewater management.

*4.2. Spanish Urban Water Management's Legal Framework*

4.2.1. Water Supply and Wastewater Treatment's State of Art

Spain entirely complies with the United Nations Resolution by guaranteeing all citizens' right to have sufficient, safe, quality, accessible and affordable water, both for personal and domestic use. Even though in some areas of Spain water is a scarce and poorly distributed resource, it has been possible to guarantee the supply through public works investments financed with European and national funds. Users and private capital also participate. That allowed mitigate crises caused by water scarcity (structural situation) and drought (short-term deficit). Supposing we were qualifying urban water management in Spain, it would obtain a remarkable grade in water supply and a fair approval in terms of sanitation and purification. Water supply and quality has been ensured, with positive advances from public and private management. However, there is still an investment deficit problem that is even more acute in sanitation infrastructures addressed with Spanish fulfilment with EU wastewater treatment legislation controversial. Thus, the coverage is close to 90% of the total concerning the pollutant load. However, it is still far from the ambitious objectives set by the EU Water Framework Directive 2000/60/CE (WFD) for purification water in municipalities with more than 10,000 inhabitants, since only 32% of these Spaniard's territories have the tertiary treatment systems as required by EU legislation [14]. As we can see, the provision of urban water services is a first-order challenge for humanity, which operates with different degrees of efficiency in each country depending on its capabilities of Development. Since water scarcity and unjust distribution, water management is associated with risks that enhance these challenges, such as legal complexity and environmental unsustainability. The Human Right to water supply and

sanitation around the world is still an asymmetric and scalable conquest. The integration of different SDG 6 framework policies is a representative strategy for addressing the major bottlenecks and threats. The European Union overview—compared to emerging countries—offers adaptable solutions for Brazilian new regime.

As established by Royal Legislative Decree 1/2001, of 20 July 2001, which approves the Spanish Water Law's revised text (therefore TRLA), water is a public good. Its use is subordinate to the general interest. The urban water supply occupies the first order of priority, by legal imperative, above other consumptive and non-consumptive uses (Art. 60). The supply of water (upper and lower phases) consists of the collection and delivery of water (adduction), treatment for human consumption (purification), transport through arteries or main pipes, storage in regulatory tanks at the head of population centers and the completion in the connections and meters of the houses. Includes domestic and urban uses; by the contrary, excludes agricultural and industrial services (except for small amounts and connected to a municipal network) [15].

With an average water consumption of 128 liters per inhabitant/day, the average price of domestic water in Spain is one of the lowest in Europe (Figure 1). It stands at around 0.9% of the total expenditure of Spanish households, significantly less than 3%, which is the UN's reference in the case of supply as a limit of affordability to guarantee humanitarian standards.

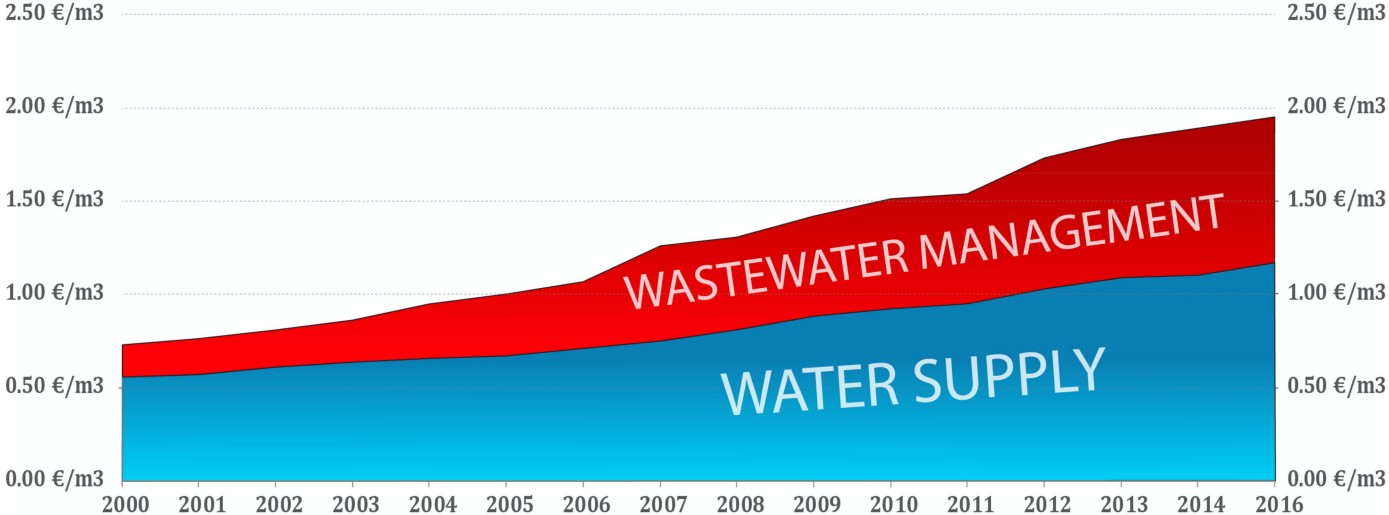

**Figure 1.** Evolution of the urban water supply (blue) and sanitation (red) price in Spain (euro/m$^3$). Source: Locken smart access solutions company, 2016 [16].

The water sector is peculiar when compared to other regulated sectors of economic activity. Different existing water management models coexist in it (direct, indirect and mixed management). The water management model in Spain serves as a reference to other countries in the world. However, the water sector's problem is complicated because an ideological perception persists where there should only be a technical analysis.

When deciding on the management model, the Local Entities exercise a discretionary power that must be motivated according to efficiency parameters. Ninety-five per cent of Spanish municipalities have less than 20,000 inhabitants. Due to the high cost of constructing treatment plants, public administrations should rethink whether competition in sanitation and treatment should continue to be managed mainly by municipalities, given the supra-local and State interest in this regard [17].

In Spain, a classic decentralized model prevails. The continuity of the service and competitiveness are more successful than the ex-post control carried out after contracting. Heterogeneous regulation sometimes prevents sufficient stability and legal certainty from

being achieved, and can hold back private investment. Nevertheless, the competition and technological support to create efficient units is a strength of the system.

One of the most severe problems is the heterogeneity of rates and the difficulty of achieving self-sufficient cost structures (cost recovery principle). In many cases, the price asymmetry is justified for technical reasons. However, in others, the indirect costs related to the service provision, such as the depreciation of the infrastructure, the financing costs or the opportunity cost, are not correctly incorporated into public accounting systems. There are differences of up to 5 percentage points between the municipalities that pay the most and the municipalities that pay the least for water in Spain. The disparity of tax figures between regions, as significantly happens with the sanitation fee, makes it difficult to carry out a homogeneous analysis. Likewise, the investment effort, more consistent in past decades, has been considerably reduced, generating a structural deficit that prevents infrastructure renovation.

4.2.2. Diversity of Models and Atomization of Urban Water Management Entities

With 0.65% of GDP, Spain's water sector has a turnover of 7654 million euros per year and offers direct and highly qualified employees to more than 32,000 people [18]. It is a dynamic and mature sector with great employability, high capacity for innovation and future. The global leadership and competitiveness of the industry's Spanish companies appear to consolidate thanks to the optimization and internationalization of processes and the development of research and development and innovation projects. However, its success is also the result of the experience of managing a traditionally scarce resource in Spain, which has forced it to perfect its forms of management [19].

According to the latest data from the sector [20], 35% of the population is managed by public entities, 33% by private companies, 22% by joint ventures and the remaining 10% by municipal services. This atomized management is organized into more than two thousand independent services, and many others delegated to supra-municipal entities. In general, it cannot be said that there is a preference for public or private management, nor that one or the other is more efficient or more appropriate under Spanish regulation. The final photo shows the operational balance between direct and indirect management modalities.

The choice of the model, i.e., the service provision, depends on each municipality's conditions to operate in a given context; and technical and political factors. Comprehensive water management following pure efficiency parameters would lead us not to think in terms of administrative boundaries, but rather to seek management formulas that take into account space, physical contour, hydraulic works, terrain orography, scale economics. In short, the territorial and demographic scope in which people live and work. The decision to choose one model or another depends on the municipality or municipalities' technical and volumetric characteristics that water services must be provided: altitude, orography, availability of the resource, government entities, number of inhabitants or technological infrastructures. Moreover, above all, the costs of providing the service or the effective cost of the service. Regulation and calculation of operating costs are essential to ensure the viability of the model and guarantee supply. The urban water cycle management model is (or, at least, should be) a political decision based on technical, economic and legal parameters.

Urban water services are provided under a quasi-monopoly regime. They must be insured and regulated by the Administration (Art. 25.2, 26 and 86.2 of Law 7/1985, of 2 April 1985, handling the Local Regime's Bases Law (therefore, LBRL). Article 106.2 of the Treaty on the Functioning of the European Union (TFEU) establishes the non-application of competition rules with the provision of these services. It considers the water supply and sanitation service as an economic service of general interest subject to environmental regulation by the EU (DMA) and water quality protection (purpose of economic, water and social efficiency) [21]. It is a functional concept that does not consider who owns the service or who provides it, nor its formal legal regime, allowing a more limited or moderate application of EU competition law concerning those activities that respond to this notion

of general interest. It caters, fundamentally, to the needs of the general welfare and its submission to obligations and rules that exempt free competition and are defined as public service or universal service obligations.

Urban water services are made up technically with different activities, although not sufficient from a physical, legal, accounting, and functional perspective. In Spain, water is in the public domain, and the regulator is the Public Administration. There are no privatizations of water as a public good; its use and exploitation is subject to a concession and authorization regime. Sanitation services are usually provided by the same operator that supplies the water. In some municipalities, management is done by private borrowing companies under a concession, joint venture or other indirect management modalities. Nevertheless, water, it is important to underline, continues to be publicly owned. The most important responsibilities, both with regulation and inspection and control of the service, correspond to the Public Administration [22].

The current regulation model of urban water services in Spain is articulated under a decentralized regime. The competencies that affect the integral urban water cycle are fragmented. The resource allocation between different users and the discharges' control depends on the competent authorities in water matters, the Basin Organizations (hydrographic confederations and autonomous organizations). The municipal government owns the water supply and sanitation services (city councils and county councils). The management of water quality depends on various administrations (Basin Organizations, Ministry of Health, Autonomous Communities or Local Administrations). The tariff and price structure depend on the municipal bodies, although the Autonomous Communities set fees for purification. In many cases, cooperation and collaboration between administrations are frequent through Agreements and administrative integration formulas that provide novel solutions and sufficient financing to develop expensive hydraulic infrastructures. And, also, through the rest of specific sectors of regulation.

The local administration, mainly, is competent to provide urban water services taking control and providing the service or contracting with an entity by indirect management. However, the power of control (whether this Administration exercises direct supervision or not) is inalienable and must be produced. This issue is often neglected due to the lack of technical capacity and specialized knowledge of the Administrations; but, also, due to a lack of interest, responsibility or personal resources as Arana García has stated: "A large criticism spilt on the breaches or irregularities of the private agents is mainly due to the absolute lack of control or cooperation on the part of the public Administrations that manage the urban water services. Why has this happened? Often, due to the lack of technical capacity and knowledge on the administrations that own the service. Clearly, in others, due to a tacit and unspeakable agreement between service providers and the responsible Administration. The Administration can relax its controls by not demanding from the private partner certain benefits or obligations in exchange for financing for the City Council on duty or as a placement agency for those related to the party in power" [23].

4.2.3. The Effective Cost and the Preference for Indirect Management

The Bases Law of the Local Regime (therefore, LBRL) establishes that domiciliary supply and sanitation services are municipal responsibility. The Spanish Administration's self-organization power allows the use of discretionary formulas to a certain extent to choose the direct and indirect management formulas that best suit its characteristics. However, the legislation establishes limitations that reduce the margin of discretion when choosing one model or another. The Art. 86.2 of Law 40/2015, of 1 October 2015, on the public sector's legal regime, establishes that direct management can only be chosen if it is a more efficient option than public procurement and is sustainable and effective, applying profitability criteria economical. In practice, this precept constitutes a principle of economic subsidiarity (preference for indirect management unless it is conclusively proven that direct management is more efficient and profitable).

In this sense, also, the wording of Art. 85.2 of the LBRL after the reform carried out by Law 27/2013, 27 December 2013, on rationalization and sustainability of the Local Administration (hereinafter, LRSAL), which establishes that: "must manage public services the most sustainable and efficient way among those listed below . . . ". The precept opts for the most efficient and sustainable formula, responding to economic criteria that avoid increasing debt, adjusting the recipe to a balanced budget. One or another's choice must be accredited and motivated in the administrative file in which they decide for a direct or indirect model.

Art. 116 *ter* of the LBRL establishes a series of transparency and accountability measures, including assessing the cost of services as a rationalization parameter. It also determines the effective cost and recovery of the expenses of the services they provide local entities. And the referral to the Ministry of Finance of the decision for the publication of this information.

The impact of costs on users defended by the OECD [24] is based on water-related services in triple financing: public transfers, tariffs and taxes. The Ministerial Order of the Ministry of Finance and Public Administrations HAP/20175/2014, of 6 November 2014, establishes the criteria for calculating the effective cost of local entities' services. The effective cost (different from the real cost of providing a local service) should serve as the basis for selecting the management model to be implemented. That unified or supra-municipal forms of provision should be chosen when the costs are lower. When the Provincial Council proves that the agreement of two or more municipalities for the integrated management of all municipal services that are coincident entails a saving of at least 10% for the total effective cost in which each city separately incurred, the coefficient of Weighting will be taken into account to facilitate an integrated management of service as set the article 124.1 of Royal Legislative Decree 2/2004, of 5 March 2004, which approves the revised text of the Regulatory Law of the Local Finances. The Art. 26.2 LBRL, after the 2013 reform, establishes that in municipalities with a population of less than 20,000 inhabitants, the supra-municipal entity (Provincial Government or an equivalent entity) will coordinate the provision of, among others, the drinking water supply service to domicile and evacuation and wastewater treatment.

When the municipality justifies before the Provincial Government that it can provide these services with an effective cost lower than that derived from the form of management proposed by the Provincial government or an equivalent entity, the municipality may assume the provision and coordination of these services. When the Provincial Government or equivalent entity provides these services, the service's effective cost will be charged to the municipalities based on its use.

The effective cost means identifying real assessment in providing the services as they are defined through the municipal budgets' information to establish some indicators of efficiency and quality for the service to a homogeneous group of municipalities standardizing the calculation methodology. The objective is aimed to eliminate administrative duplications and reducing public spending. The approval of the management formulas' limitations has raised conflicts related to local autonomy (Spanish Constitutional Court Sentences 41/2016, of 3 March 2016, and 111/2016, of 9 June 2016). In general, experience shows that public administrations are unwilling to assume powers if a budget increase does not accompany them.

Unfortunately, one management model's choice or another has been influenced by ideological conceptions, according to which its supporters have chosen to defend the alleged advantages of one management model over the other. Arguments in favor of indirect management mean a more specialized companies' know-how and advanced knowledge that would generate an extensive and diversified management experience. It is also value:

- The cost savings that may come from having to achieve contractually predefined financial profitability targets;

- The greater transparency, by having to comply with the rules that govern for commercial companies;
- The more significant raising of private capital to finance infrastructure, which results in less financial responsibility for the State;
- The fact that it can help reduce political interference and make operational plans to develop longer-term approaches, regardless of political times;
- Or, finally, that the Administration continue to reserve, as the service owner, the ability to oversight and demand contractual compliance with the agreed requirements by not taking a party position.

On the contrary, indirect management would consolidate -with no control- a natural monopoly in which large lobbies and business groups share the market. Measures can also increase collection in citizens' invoices due to the need to increase commercial margins, or small and medium-sized companies' participation can be reduced at the local level.

*4.3. Urban Water Control: From Responsibility to Transparency*

The water administration's transparency requires that, regardless of the chosen model, public or private, the decision be sufficiently motivated according to technical, economic, and financial studies that demonstrate and accredit, in reliable terms, its suitability. Thus, political control is a decisive factor in the decision's behavior, and, as such, it deserves to be analyzed [25]. Public-private participation, usually is articulated through a public contract that defines the effectiveness of the provision to be made and the terms in which the legal relationship must be developed. The degree of detail with which the obligations are drawn up in the contractual specifications may condition the provision's purpose [26].

The EU Directives on public procurement have introduced variations with the powers attributed to the "contracting body" (before contracting) and the "person responsible for the contract" (after contracting the service), or the new control system and supervision of public procurement, all of them of great importance and complexity, which we cannot go into at this time. We want to mean that the Administration has a reviewing power as the service owner and must assert its powers of inspection and control towards the service providers, regardless of whether the legal relationship is based on a principle of legitimate confidence. This verification can be carried out both in regulation and provision, through essential documents to support the legal relationship: service regulations, administrative specifications, Supply Master Plans, financial studies, rate policies, annual reports and audits, instruments of public information and citizen participation, etc. Efficient management must promote competition, define the obligations through technical reports, clarify the parties' functions and rights, and grant due to predictability and legal certainty, describing the service's operational risk issues.

For European regulation, the management formula chosen for the provision of public services is indifferent, as established by Directive 2014/23/EU of the European Parliament, of 26 February 2014, on contracts concession. Nevertheless, the need to promote public-private collaboration to finance infrastructures and hydraulic systems (construction, renovation, management and maintenance) is recognized; and as a mechanism, to attract technological knowledge and capacities in the field of research, development and innovation.

Law on Public Contracts 9/2017, 8 November 2017, integrates the EU Directives' content on public contracts and places the accent on mixed economy companies. A public operator (Administration) participates and a private operator (technological partner) as a successful management model [27].

The process of integrating public-private collaboration also materializes in the reduction of contractual figures and administrative simplification in order to reduce indirect management contracts to two specific types: concession contracts and service contracts, depending on the existence or non-transfer of the "risk of providing operation" or "operational risk", as derived from the transposition of the package of Community Directives on procurement, defined as that which is beyond the control of the parties and occurs as a consequence of the uncertainties of the market. It consists of a demand risk or a supply risk,

which capitalizes the net present value of all the concessionaire's investments, costs, and income. This operational risk is also linked to factors of competition, insolvency of debtors, insufficient income, or derived from liability for damages caused by an irregularity in the provision of the service. It consists of a demand risk or a supply risk, which capitalizes the net present value of all the dealer's investments, costs, and income. This operational risk is also linked to factors of competition, debtor insolvency, insufficient income, or derived from liability for damages caused by an irregularity in the provision of the service [28].

It is raised whether it is possible to redeem a concession to provide the service directly when provided indirectly. In any case, the concession rescue is a discretionary decision that can only be adopted "if there is a prevailing public interest mediating compensation in favor of the concessionaire for the damages that said agreement causes it" (Spanish Council State Opinion No. 2918/2003).

The concession rescue is a cause for termination of the contract, not due to contractual breach attributable to the contractor, but based on a cause of public interest. A confiscation implies an obligation to compensate the contractor according to a discretionary decision under control by the Courts, requiring a motivation and economic rationale.

In broad, progress has been made towards a disappearance of the bailout as a prerogative of the public Administration. Furthermore, when this happens, it tends to materialize as part of an administrative liability system that requires expropriation and financial compensation.

### 4.4. Water and Sanitation as a Strategy of Public Health and Democratization in Brazil. What Would We Learn from Spain?

#### 4.4.1. Wastewater Treatment: A Recovery Master Plan for Leaving Behind the Covid-19 Crisis

Urban water management is linked with global phenomena and changes that could affect to the resource availability. Among them, it is worth mentioning the climatic change and the fulfilment of the principles that govern the management of the resource: water economy, efficiency, management unit, water unit, respect to hydraulic systems, environmental sustainability, restoration of nature, cost recovery, demand satisfaction, regional balance and development, etc. Similarly, technological transformations, significant investments and the development of concepts such as circular economy, water quality and ecological transition are predicted.

Basic sanitation is a pillar to improve people's lives and the environment in which they live, being one of the leading indicators of a country's quality and social and economic well-being. Under the United Nations' supervision, developed countries have been struggling to develop programs to improve water and sanitary conditions, which are very precarious. International cooperation, based on the creation of backgrounds and contingency networks, as well as the provision of technical assistance to improve the living conditions of affected communities through approaches such as "AAAQ-criteria" or "4A-criteria ", which allows to develop a framework of action based on availability, accessibility, affordability, acceptability and quality of access to water and sanitation.

Basic sanitation of water is a prerequisite for achieving human rights. However, it also requires to implement tangible public policies in housing, education, equality, economic, etc. However, securing basic sanitation for populations is a challenge in emerging countries like Brazil, since access to drinking water and sanitary sewage treatment prove to be costly services, and cannot always be provided by the municipalities, in a way that adds incompetence in public management, social inequality and the growth of environmental degradation.

In Brazil, at least 83.5% of the population has drinking water supply, and 52.4% has collection of sanitary sewage, with only 73.7% of the sewage being treated with conditions to return to the system [29]. This percentage is much higher than that registered in the countries of the European Union, which allows European administrations to focus on developing other public policies, such as the reuse of water and organic matter, energy and climate change, reduction of impacts and conditions that may have other sectoral policies such as urban regulations and waste disposal.

The new Brazilian Federal Law No. 14,026/20 aims to encourage the investment strategy by promoting public-private collaboration formulas, making the system economically viable even in regions that may be less profitable due to their population or technical characteristics or accessible to the provide this service. The role of the State, in this sense, is aimed at promoting business investment in areas that are less attractive for investment by large corporations in the water sector. The Brazilian model, territorially dispersed, but with large megacities that absorb most of the country's population and services, aims to redesign the formulas for providing these services to encourage investment and establish better transparency accountability mechanisms in public management [30].

Thus, water sanitation can be a development and recovery lever in the exit strategy from the Covid-19 crisis. The European water sector has recently made proposals to materialize investment in the urban water cycle's specific projects, totaling 13,775 million euros. Not only that, sanitation policies are related to values of any democratic society such as the Right to life-hygiene and the environment, but other global strategies such as climate change mitigation or the fight against the pandemic are also connected. For example, these are reflected in the application of technologies that minimize the carbon footprint, the sampling that has been carried out to control the coronavirus in large cities, or the strengthening of the European Circular Economic Strategy. In Spain, for example, more than two-thirds of the operators in the water sector have energy use devices and have plans to control the carbon footprint [18].

Therefore, it is not surprising that the United Nations General Assembly has designated the Resolution of 25 September 2015 (A/RES/70/1) the sanitation service as one of the new 2030 Agenda's central axes. Concerning the Brazilian context, sanitation appears as goal number 6 of the SDGs, being undoubtedly one of the most critical targets.

### 4.4.2. Wastewater Treatment and Policies in Europe

Water Sanitation is a topic of a greater importance, given its environmental and hygienic-sanitary imprint, as well as to the preservation of the quality of the waters and the associated ecosystems. The assurance of its provision, the control of public-private participation, and economic-financial management are the main lines of action.

In the 90s of the last century, The European Union published two necessary Directives on sanitation [31,32]: first, Directive 91/271/EEC, of 21 May 1991, concerning urban wastewater treatment (DUWT), under which obligations were established aimed at Member States consistent in adopting the appropriate measures to ensure that wastewater was adequately treated in a phase before its discharge; and, secondly, the Water Framework Directive (WFD), which establishes the Community framework for action in the field of water policy, which imposes the achievement of objectives environmental quality and constitutes the leading standard of European water law and the most critical measures in the ecological field [33].

Focusing on the DUWT, it was transposed into the Spanish Regulation by Royal Decree-Law 11/1995, 28 December 1995, which establishes the rules applicable to urban wastewater treatment. With its transposition, a technical criterion of great relevance is introduced: the number of inhabitants-equivalents, a concept that defines the polluting load of both people, animals and industries, and urban agglomerations, which are the areas that present a sufficient concentration to collect and conduct wastewater independently. Likewise, the standard clarifies concepts such as urban, domestic or industrial wastewater.

The Directive established a set of obligations to which it conferred progressivity and greater or lesser flexibility over time (differentiating the groups according to whether they had between 2000–10,000 hectares, more than 10,000 hectares, or more than 15,000 hectares), requiring the provision of systems of collectors, the condition of WWTP with "secondary treatment" and the provision of WWTP with "adequate treatment". Finally, it established prior regulations or specific authorizations for all discharges into the natural environment from urban wastewater treatment plants and the agri-food industry's facilities and industrial wastewater discharge to urban wastewater collection treatment systems.

The urban wastewater law was developed in Spain by Royal Decree 509/1996, of 15 March 1996 (Table 3). Its execution and possible development corresponding to the Autonomous Communities, which have defined, correlatively, the municipal powers, in the cases in which they have developed it.

**Table 3.** European Union regulation and its implementation in Spain.

| Field | Basic EU Regulation | Spanish Regulation Implementation |
|---|---|---|
| Comprehensive water protection | Directive 2000/60/CE | RDLeg. 1/2001 (TRLA) RD 849/1986 (RDPH) O. ARM 2656/2008 (IPH) River Basin Management Plans (RBMP) RD 817/2015, (RDE) |
| Urban wastewater and sensitive areas | Directive 91/271/CE | RD 509/1996 RDL 11/1995 |
| Vulnerable areas and affected waters | Directive 91/271/CE | RD 261/1996 |
| Integrated pollution control | Directive 2010/75/UE | RDL 1/2016 |

Despite the transposition of the DUWT into the Spanish legal system and the approval by the Spanish State of plans to carry out the EU Directive (see *ut. infra*), the Kingdom of Spain has been condemned by the Court of Justice of the European Union in several pronouncements for not complying with the Directives. The reasons have been diverse, such as not having treated the water coming from urban agglomerations with a population of more than 20,000 equivalent inhabitants or the lack of construction of treatment plants in urban agglomerations with more than 15,000 inhabitants (see the 2018 Sentence of the European Court of Justice, C-205/17).

The mandatory full purification of waters implies that the Kingdom of Spain through National Administration faces the liability from the European institutions in non-compliance. The lack of definition with the responsible authority has caused failures in the responsibility system when, for example, the Autonomous Communities have decided not to intervene by regulating or managing the service. In general terms, the design of responsibilities is not easy to define, neither according to the establishment of a sanitation fee that makes the principle of solidarity effective (the cost of treatment plants is variable depending on many parameters), nor in regarding the distribution of financial obligations [34].

According to recent data, of the 2100 agglomerations of more than 2000 inhabitants that must purify their waters in Spain, about 550 are in non-compliance [35] (Figure 2). The problem is rooted in the dispersion of powers and the municipalities' limited financial capacity to undertake these works, which can be very expensive.

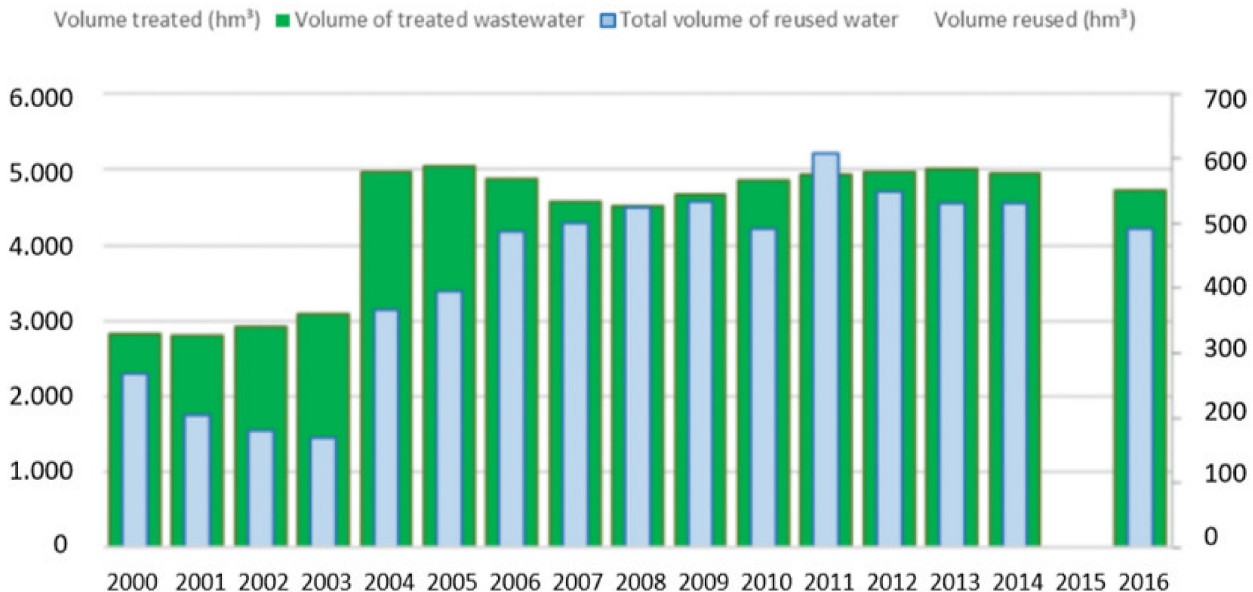

**Figure 2.** Evolution of the volume of wastewater collection (green) and treated water reuse (blue) in Spain. Source: Spanish DSEAR Plan draft, 2020.

### 4.4.3. Wastewater, Sewage, Purification and Sanitation's Public Services: The Difficult Separation and the Problems of Management that this Entails

The sanitation service encompasses actions that include the collection and conduction of the water, its treatment, and its return to the environment. These technically differentiated processes aim to eliminate or reduce the waters' pollutant load before they are discharged into the receiving environment, the public channels. As a service of the urban water cycle, the collection of urban and rainwater wastewater from population centers through municipal sewerage networks to the point of interception with the general collectors or the collection point for treatment, depends on a complex infrastructure network that requires a large investment.

Wastewater can contain untreated substances like nutrients (nitrogen and phosphorus); solids (including organic matter); pathogens (including bacteria, viruses, and protozoa); helminths (intestinal worms and worm-like parasites); oils and fats; runoff from streets, parking lots and roofs; heavy metals (including mercury, cadmium, lead, chromium, copper) and many toxic chemicals, including PCBs, PAHs, dioxins, furans, pesticides, phenols, and chlorinated organics.

The terms 'wastewater' and 'sewage' are regularly used interchangeably, however there are differences between both. In fact, 'sewage' is considered a subset of wastewater [36]. Wastewater is water that has suffered anthropogenic influence and has deteriorated its quality. It includes liquid waste from residences, businesses, industries and agriculture and can encompass a wide range of pollutants. Sewage is the water contaminates with faeces or urine, although it is also often used to refer to any wastewater, meaning the water that is conducted through pipes and sewage systems. On the other hand, sanitation refers to the set of sanitary measures and actions to prevent and avoid health hazards.

Wastewater treatment is aimed at improving the environmental quality of public waters [37]. It consists of collecting urban and rainwater wastewater from urban population centers through municipal sewage networks to the point of interception with the general collectors or to the end of collection for treatment or purification. Thus, wastewater sanitation is made up of two phases: low sanitation (purification process by which the water reaches an acceptable or sufficient environmental quality standard per the regulations) and high sanitation (consisting of its return to the environment once the waters have been purified) [38]. Thus, sanitation, as a network system, depends on other processes: the purification of wastewater (through a process of interception and transport across general

collectors, its treatment and the discharge of the effluent to continental or maritime water bodies), the management of the spill (the administrative authorization where the conditions are established in which the treated water must be returned to the environment) [39] and the reuse/regeneration of water (through which the water is reused, once purified, in cases where to be determined, being destined for other uses such as garden irrigation or street washing) [40]. Purification is the prelude to reuse, so when speaking of sanitation and water treatment, we would not be referring so much to the management of a waste, but the use of an authentic resource [41]. However, water reuse is still an underused resource due to social resistance and demanding sanitary regulations.

Sanitation can be carried out through unitary or separate networks. The former is designed and built to receive a single conduit, mixing both the wastewater (urban and industrial) and the rainwater generated in the metropolitan area covered by the network. On the other hand, the separative networks introduce two independent channels: the sanitary sewer system and, the other, the storm sewer system. The separative network is ideal, since it reduces purification costs and simplifies processes, since the treated flow is lower and more constant; it also allows transporting a lower polluting load. For the proper functioning of the separative networks, there must be a control of discharges and diffuse points of contamination and the duplication of downspouts in building installations (using technical standards in building matters). The inspection and monitoring of the same, and the cleaning and conservation, the control of illegal dumping, are decisive elements of action, which must be helped by the correct technical design. So the networks of infrastructures must have an adequate hydraulic constructing design, with a suitable slope, avoiding siphons and installing cesspools by doing the appropriate maintenance and conservation work.

For example, a pipe inclined according to a correct velocity is considered to be one in which the water moves at a speed of 0.46 m/s. At a lower rate, solid matter can tend to settle and clog the conduits. Monitoring of the same, and the cleaning and conservation, the control of illegal dumping, are decisive elements of action, which must be helped by the correct technical design. Among the main problems that sewage networks support are: obstruction by roots, damage by works, aggressive spills, or the reduction of water capacity due to the block of substances, mixtures and, remains of textiles or wipes [42]. This raises the cost of maintenance work and damage to infrastructures, with inspection and cleaning operations critical to prevent and reduce these damages in their different technological manifestations.

Most of the Spanish municipalities have organized and managed purification in a way other than supply. As a legal activity, water Sanitation is challenging to conceptualize due to the proliferation of rules regulating it [43]. Some use the terms "sanitation" and "purification" interchangeably, referring in some cases to sanitation (Art. 25.2 LBRL) and in others to "purification" (86.3 LBRL) as equivalent [44].

The difference between high and low sanitation in the Spanish system can become significant due to the competence problems that arise, because, although the sewerage is a municipal responsibility, the collectors or conduits that collect and conduct urban wastewater from the Sewerage networks to the treatment station are regional competence if their legislation so provides. Also, more and more, to supply and sanitation, the activity of reusing water for purposes other than human consumption is added, making it possible that the water can be used, once it has been purified, being destined for other uses (irrigation of gardens, street washing). Despite everything, it is, to this day, a resource that is underused due to the reluctance of society and the psychosocial component that its added uses entail.

Notwithstanding the lack of definition of competencies is the difficulty of separating the technical processes that negatively affect the liability system. In this way, the State can intervene to dictate primary legislation on the basis and general coordination of health (Art. 149.1.16 CE), the bases of the local action regime (Art. 149.1.18 CE), legislation, management and concession of resources and hydraulic utilization when the waters concur

in more than one Autonomous Community (Art. 149.1.22 CE) and the necessary legislation on environmental protection (Art. 149.1.23 CE), among others.

Secondly, the Autonomous Communities can develop the norms on environmental matters (Art. 148.1.9 CE) and the primary state legislation, being able to assume competences in the projects, construction and exploitation of the hydraulic uses, channels and irrigation of interest of the Autonomous Community (Art. 148.1.10 CE). The Autonomous Communities have great responsibility regarding the determination of urban agglomerations and establishing a representative body of the municipalities located there. The financing and construction mechanisms of wastewater treatment plants (WWTP) are created by public entities, generally at the regional level, and administer and control regional systems, contracting the management (operation and maintenance) to private companies. The Autonomous Communities have created Entities dedicated to the purification and sanitation of wastewater, creating their taxes known as "sanitation canon". The Autonomous Communities have issued their regulations on sanitation, developing state legislation, and defining municipal powers in this area.

Municipal intervention is, in third place, decisive in the matter of sanitation. The competences in favor of the municipalities are not prescribed in the water legislation, but in other special laws (Art. 42 Law 14/1986, 25 April 1986, General Health) and in Law 7/1985, of 2 April 1985, of Bases of the local regime (Arts. 25 and 26). In what interests us now, Art. 25.2.c) of the LBRL indicates that the municipalities have the competence to evacuate and treat wastewater. Therefore, it should be noted that Local Entities can issue regulations and ordinances to regulate the sanitation service through, for example, disposal ordinances. The Art. 26.1.a) of LBRL determines the obligation that the sewerage service be provided by the municipalities, increasing their legal obligations as the population increases.

Finally, it is possible, increasingly used, that different municipalities are grouped to provide this service through associative legal forms that make up supra-municipal entities under the Provincial Councils' protection and coordination. In these cases, different organizational techniques are used that go through the creation of specific Entities and that use forms of direct management, concession, concert, association, counties, metropolitan areas, consortium or supra-municipal entities. By following this idea of supra-municipality of the service, the Autonomous Communities and the Provincial Councils (within the province) assume greater management responsibilities to comply with European commitments. Urban agglomerations, metropolitan areas, but also small municipalities and nuclei of the dispersed population (think, for example, of the problems that occur in "emptied Spain") make it necessary to review the traditional concepts of competition and solidarity by how, more and more, the administrative limit between municipalities is, and that is the key, more diffuse, as regards the management of this competence.

The lack of investment in the wastewater system adds to each political entity's attributions. It is inserted as one of the most accusing problem, serving Spain's experience (lack of investment to maintain and expand the sanitation infrastructures) as a significant warning for Brazil's basic sanitation system.

### 4.4.4. The Problems Linked to the Lack of Investment

The characteristics of the urban water cycle make that they cannot be provided by small municipalities, so it is common for different administrations to join together to provide the service and save on infrastructure costs (treatment plants, desalination plants, constructing collectors). It should note how the local Administration does not have the resources to undertake the demanded infrastructures. In some cases, the declaration of "public interest infrastructures" has allowed the State to finance works of the based on political criteria. The financing of sewage treatment plants, constitute a singular modality of public works (Art. 122 TRLA), it has been mainly autonomous or State, and, when the construction of WWTPs has been contracted, the local entities have not always been contracting authorities.

The administrations involved in sanitation competences must materialize them by proceeding to the adequate wastewater: by carrying out planning, equipping and sanitation actions [37]. Among them, planning has a remarkable impact, either through the National Hydrological Plan, for the whole of the State, which identifies in its Annexes a good number of sanitation infrastructures by which they are declared of general interest. Or through the river basin management plans, for each of the river basin demarcations, which, through the specific program of measures, establish the environmental quality objectives and list the necessary infrastructures required for this purpose (Art. 42.g) o') TRLA). However, it will be, above all, the specific sanitation plans that will establish the most relevant actions in this matter, such as the National Plan for Sanitation and Wastewater Treatment, approved on 17 February 1995, by the Council of Ministers; the National Water Quality Plan (2007–2015), and, more recently, the currently draft-processing National Plan for Purification, Sanitation, Efficiency, Savings and Reuse (DSEAR Plan in Spanish) which aims to achieve greater efficiency in the areas of purification, sanitation and reuse of reclaimed wastewater, improving the approach of river basin management plans and other regulations. The DSEAR Plan draft, in brief, considers seven highlights:

1. Define criteria to prioritize the measures defined in hydrological planning.
2. Strengthen cooperation between public administrations.
3. Improve the definition of actions that should be considered of general interest of the State.
4. Improve the energy and comprehensive efficiency of the purification and reuse plants.
5. Improve the financing mechanisms of the measures.
6. Promote the reuse of wastewater.
7. Promote innovation and technology transfer in the sector of the water.

In any case, sanitation planning corresponds mainly to the Autonomous Communities, which can also legislate through other sectoral powers such as the environment, spatial planning and urban planning.

Sanitation systems require a large investment and must be financed according to a specific economic-financial regime. In Spain's case, community policies have contributed generously, over the years, to the construction of technical infrastructures such as treatment plants and desalination plants. On the other hand, public-private participation mechanisms have been generated through different contractual formulas that assume, from the infrastructure's design and construction, to publicly-owned service management. Likewise, finalist taxes have been created, such as the sanitation fee, which have contributed to sustaining the system. Sanitation requires a large investment that does not always have sufficient political return due to the costs of construction, renovation and maintenance of infrastructure.

Currently, Spain has more than 165,000 km of sewerage network (3.5 m per inhabitant) and 460 storm tanks and 2300 wastewater stations (WWTP) that purify an annual volume of 3769 hm$^3$, about 222 liters of water per inhabitant per day. Nevertheless, the water sector faces financial difficulties that require an economic boost to address the lack of investment dragging on in recent years, forcing us to rethink the situation (Figure 3).

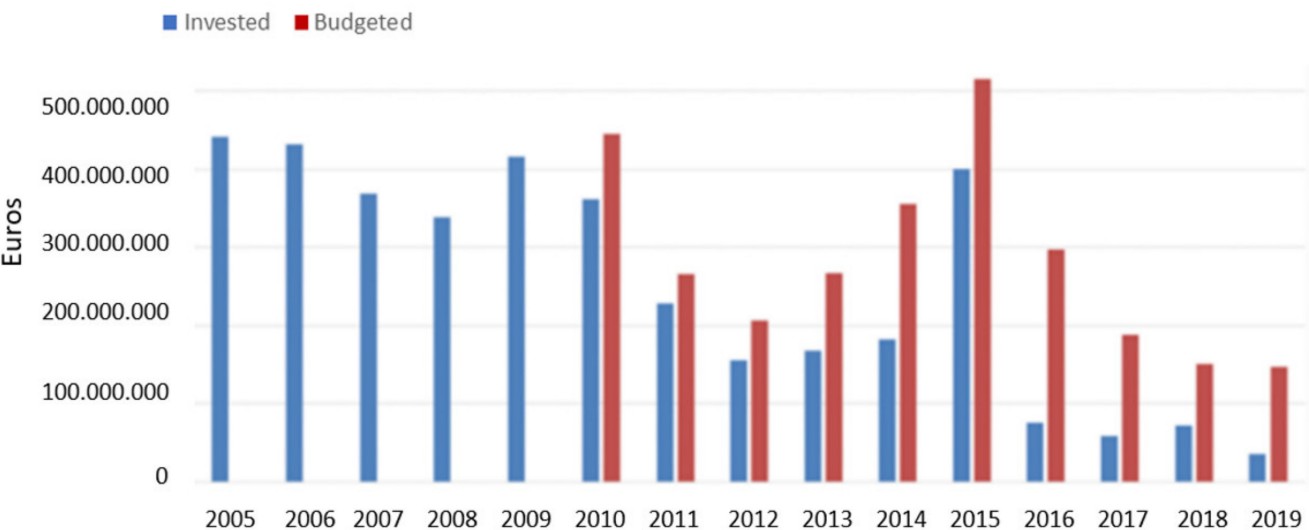

**Figure 3.** General investments in sanitation and purification (blue) and budgeted amount (red) from 2005 to 2019. Source. Spanish General Water Directorate of the Ministry of Spain-DSEAR Plan draft, 2020.

It insists on the need to approve a Law on the Urban Water Cycle that provides greater legal security for investments or the need to create an independent regulatory authority, in the style of what other neighboring countries have done.

The reduction in Spain's investments is getting more severe than the rest of the eurozone countries. The low investment made and the planned one compromise the economic viability and means that the rates do not cover the real costs and only cover the urban water systems' operating expenses. The accumulated investment deficit increases, placing the renewal rate of sewerage networks at around 0.38%, significantly less than the 2% per year, which is estimated as necessary. The XVI National Study on drinking water and sanitation [18] puts this investment deficit at 3157 million euros per year. The Report considers it is necessary to invest 1900 million euros in annual work investment, when 479 million euros are invested. Moreover, in 2221 million euros of infrastructure renovation investment, only 555 million euros are being invested.

The evolution of historical investments in Spain's water infrastructures has fallen from 0.36% of GDP (2007–2009) to 0.14% of GDP in the 2014–2017 period. It contrasts with the situation in the European context countries such as Germany, France, UK, Italy or the Netherlands, where the trend has been to the upside (Figure 4).

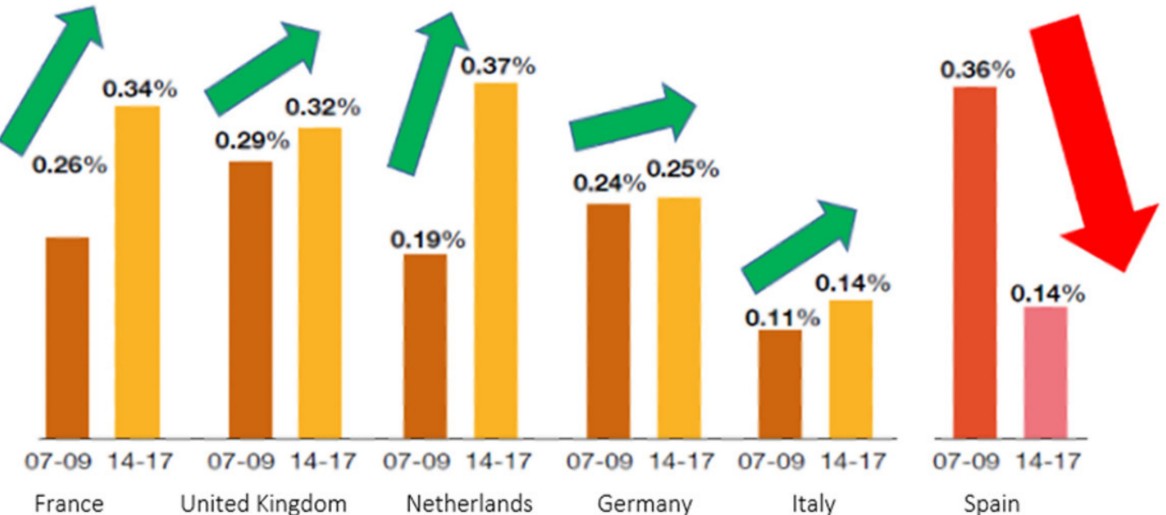

**Figure 4.** Evolution of historical investments in urban water infrastructures by country (% GDP). Data sources: GWI Eurostat, Spanish Ministry of Development, AEAS [18].

Insufficient cost coverage is compounded because the price of water in Spain is significantly lower than the European average €7.4 (€88.77 a year) by domestic supply and sanitation services, with a substantial heterogeneity rate [18]. Technical reports also criticized the spread of jurisdictional responsibilities in deteriorating facilities and services [45].

Some proposals aim to add urban water systems to achieve economies of scale and develop the companies' technical and management capacities that provide the service. As the doctrine points out, this phenomenon must be based on a progressive "pooling of service". The large number of operators (more than 2000 in a territory of 8124 municipalities) turns the Spanish model into a system with "decentralized, heterogeneous and variable regulation in the national territory" [46]. When choosing the management formula (direct or indirect), the traditional administrative discretion tends to be more and more motivated.

As we indicated (see *ut supra*), at the local level, local competence's public services must work most sustainably and efficiently (Art. 85.2 LBRL), which response to no increases in debt and the possibility of developing a balanced budget. At the state level, Art. 86.2 of Law 40/2015, 1 October 2015, indicates that direct management can only be chosen if it is a more efficient option than public procurement and is sustainable and effective, applying economic profitability criteria. In practice, this supposes a principle of economic subsidiarity (preference for indirect management unless it is reliably proven that direct management is more efficient and profitable).

On the other hand, there is talk of the need to harmonize tariff criteria and converge in water prices, through the establishment of transparent tariffs and a correct application of the principle of cost recovery, as stated the European Court of Justice by indicating that the recovery of costs established in Art. 9 of the Water Framework Directive "does not impose a generalized obligation to rate all activities related to water" [47].

The notable differences in water rates, above 300% when comparing some municipalities with others, the highest between Europe regions, have an economic impact on users and can affect competition, solidarity between areas, and the public-private collaboration model's very competitiveness. Despite, these price differences are not always disproportionate, as they are usually criticized, and respond instead to the technical conditions in which the service is provided, so an operational analysis must be carried out in reasoned terms. To this issue, the Spanish Constitutional Court Sentence no. 85/2013, amends the authorization established by Water Law 1/2006, of the Basque Country region, to determine the concepts to be included in water use pricing in urban services violating local autonomy.

Considering the financial field, a representative discussion emerges in terms of basic sanitation and wastewater in Brazil. It is about the opportunity to create a managing entity or independent institution entity as a necessary step to get neutrality and contribute to generate economies of scale. Through the Spanish case, Brazil´s case shows that a more decentralized model could be successful work in an operative way. In the short term, the Brazilian legal system (according to the new law) would encourage new investments by creating an independent authority that controls the process form a political and geographic point of view. In this context, a glace to other European regulatory models can help reach a more holistic perspective about the suitability to create a specific authority as many of them have done, as we explain.

4.4.5. Other Regulations in Europe: The Independent Authority as an Element of Standardization of the Service

Under the modern idea of the integral water cycle, there is a risk that there is hidden a bidding criterion that harms competition by attributing urban services to the company that has traditionally provided them in that population. In the last years, there have been significant political and social movements favoring the return to municipal water services management in cities such as Paris or Buenos Aires. However, most studies, national and international, insist on the idea that determining per se a greater effectiveness of what is public or private is a myth. From an efficiency point of view, the public or private nature of the operator is irrelevant. Regulation and the institutional framework are the determining factors for achieving efficiency.

A more centralized management model could be promoted, attending to specific objectives and achievements such as economic harmonization and the generation of service and control standards; or the imposition of new legal and financing forms. For others, decentralization is positive and shows the ability of the model to adapt to the local environment and context, being relevant to establish a homogeneous regulatory framework, considering that the creation of an independent regulatory entity is not necessary since planning and basin organizations they can adequately fulfil the functions assigned to this hypothetical regulatory entity.

Among the centralization supporters, there is discussion about the need to establish a unifying and independent regulatory entity (robust regulation model) or to create a more flexible institutional model through a Water Observatory (sunshine regulation). The 2015 Lisbon Charter, drawn up by the International Water Association (IWA), defines the term "regulatory or regulatory authority" as "the public authority responsible for applying and enforcing standards, criteria, norms and procedures, which have been politically, legally or contractually adopted, exercising autonomy in its control over the services, in its supervisory capacity).

There are varied examples at the European level that advocate a differentiated management of the urban water cycle. Most of them have an independent regulatory entity that performs functions aimed at controlling service provision, such as monitoring a national strategy for regulated sectors; the promotion of clear rules to guarantee legal security in the industry; the production and making available to all interested parties of reliable information; the commitment to technological innovation and transparency; the application of inspection and control mechanisms in relation, mainly, to the contracting conditions and the relationship of the borrowing entities with the consumers (access to the service, quality of the service, service rates and others).

Broadly speaking, there are two reference models in Europe: the English model and the French model. In the English model, first of all, the provision of urban water services is regulated mainly through private companies, both from a property and system operation point of view, for which regulatory agencies are created to control the rates applied, the quality and satisfaction of the service users, the cost recovery mechanisms, etc. Second, the French model is characterized by establishing a long-term legal relationship between the government of a territory and a service provider entity through which a "regulation by contract" is set. The price of the water is negotiated execution of the service delegated.

In England, the Administration continues to be responsible for the public water service through public agencies that control environmental decisions, the quality of the water supplied. The interaction with these private companies' users takes the economic regulator, OFWAT, which is concerned that private companies are sustainable by recovering their costs. On the other hand, it deals with assuring the service they provide to users is efficient and adequate, protecting users from the monopoly of companies and the Public Administration's political ambitions [48].

In Portugal, the *Entidade Reguladora dos Serviços de Águas e Resíduos*, (ERSAR in Portuguese, created by Decree-Law No. 277/2009, integrates water and waste management. As an independent administrative entity with regulation and supervision functions, it has broad autonomy of management, administrative, financial and its assets, being attached to the ministry with powers in environmental matters, but not subject to government supervision or surveillance in their exercise functions. This model is based on two levels, the regulation of the sector; and the regulation of the managing entities' particular behaviour.

The Italian Regulatory Authority for Electric Power, Gas and the Water System (previously called AEEGSI, from 2018, ARERA) guarantees the promotion of competition and efficiency. It aims to promote standards of quality of service. The entity is self-financed and performs meaningful regulatory, oversight, enforcement and advisory functions. In 2011, a national referendum was held in which it was decided to undertake, among other issues, the renovation of infrastructures above 65,000 million euros (30 years).

Given the European experience, we consider it interesting to introduce the debate on the suitability of creating a unifying and independent regulatory entity in Brazil. It is a complex mechanism but one that can provide legal certainty and stability. Their integration does not have to replace or displace the public service holders (this is the case in countries like England, Wales, Chile). On the contrary, the independent authority can take regulatory action on tariffs or seek financing mechanisms to work based on efficiency and scale economies. Faced with a purer model of independent power, one could also opt for a more flexible model, such as the Netherlands or Portugal, proven effective.

Among the competencies that this hypothetical independent regulator could develop, are some as relevant as technical intermediation in contracting the service, developing technical regulations (service orders), controlling public transparency and the rate regime, etc.

## 5. Conclusions

The research emphasizes public and private partnership of urban water management, adapting the regulatory and institutional structure by focusing on basic water sanitation service.

Water policies become involved in a continental country as Brazil is in terms of territory. In this context, it is needed to consider economic, social, environmental, and geographical realities. In the meantime, the Spanish model allows us to draw some perspectives that could be problematic in the Brazilian facts, for example, even though there is no decentralized regime in Brazil, a new legislative configuration of its "super" powers for ANA, or, in a practically centralized way.

The necessary wastewater and sanitation policies of a unique state entity (which has enormous responsibilities in managing the Brazilian national water resource) could be rethink. The new regulation gives the National Agency—ANA—power to redefine investments, define the contracting model (technical and mandatory criteria), stimulate cooperation between the federative entities, and reassess the expansion and universalization of two public coverage services basic sanitation endorsing operations of private companies. The "new" Brazilian model reports a centralized basic sanitation policy in a unique administration that is already overloaded with other competencies, creating a gloomy outlook for the new regime's succession. The River Basins Authorities themselves are disregarded, not the new model, not what they have, for example, water for primary sanitation purposes, a definition of a tariff model and the transparency rules for the sanitation management operation. The need to guarantee basic sanitation on the part of the State is to create economic and financial stimuli through a safe and stable regulation on public procurement that limits operational risk assumptions.

The Spanish population is satisfied with urban water management. Wastewater situation worries due to the need to face the investment deficit. It is necessary to create financial structures that guarantee water's social function while ensuring its affordability and economic viability. Among the Spanish model's advantages are the operators' capacity and technical solvency to face short, medium and long-term projects. On the contrary, deficiencies have been detected (competency confusion that can lead to the liability system failures, lack of control by the Public Administration, politicization of management or investment deficits, meanly). These shortcomings illustrate what the Brazilian legislator must avoid in implementing and developing the new Brazilian Law.

In both countries Brazil and Spain, we find a dispersed model of regulation. The different quality of the water, the more or less dispersed urbanization of the residential centers, the seasonal population in tourist areas, the disparity of autonomous fees, the age of the supply and sanitation networks, the existence of economies of scale or the way of finance the facilities, favor the integration of supra-municipal management models. Given the difficulties of developing the new Brazilian regulatory framework, we believe that it would be interesting to work on a more specific regulation on urban water management, which integrates and simplifies the environmental and economic obligations, which introduces the commitment to plan activity and encourage public and private participation.

This regulation could also generate an accounting structure to guarantee the use of sustainable technologies and the generation of new ways of financing, providing neutrality and rationality in economic, technical and social terms.

The Spanish model works appropriately—it is always possible to improve—with not an independent regulatory authority in urban water services. This possibility is valid because it works with significant regulatory, planning, and financial instruments. A Spanish weakness is focused on the relaxation in the concessionaires' controls. However, Spanish wastewater policies depend on the European Union's community policies. From an environmental point of view (in terms of water quality, sectoral approaches, specific environmental directives) and economically (as general economic interest service notion and its submission to the directive's of public contract package).

The concessions by indirect management without any control is a potential target for corruption and undue enrichment. We believe in creating an independent agency that could be feasible as an autonomy and rationalization of competence in Brazil's case. The solutions and strategies for managing public water services in cities lead to control contamination and public deficit and establish an economic-financial framework that guarantees investment. Creating an independent regulator could provide technical solutions to reduce the political burden of decision-making, working based on a standard regulation and discharging tasks from the National Water Agency, overlapping with competences. Nevertheless, performing other functions: Evaluating and controlling the performance of operators; Using regulatory powers; Resolving disputes; Standardizing technical and contractual aspects; Promoting research projects; Advising the government; Coordinating administrations with interest groups; Working in the government and transparency of the sector

Brazil's independent regulatory agency would also foster competitiveness and public and private collaboration under the service's shared supervision, but approving the know-how and knowledge can bring the private sector to a dynamic and primary environment for society's interests.

Another Brazilian problem is the scenario and open space for corruption and the lack of guarantee to fulfil with water Human Rights. The paper aims to highlight two urban water management mistakes in Spain and their possible usefulness for Brazil institutions. First, avoid political criteria interference to decide the urban water management model in each local government instead of attending economic and technical measures. An effort should make to not determine the model according to ideological approaches by managing the information with transparency to prevent the decision from behaving like an arcane of statistical, accounting and budgetary data that do not transcend public opinion. Second, facing the severe deficit of structural investment in infrastructure and redesign the financial and budgetary regime.

Brazilian Federal Law No. 14,026/20 should mark an inflexion point facing the future. To this purpose, we have carried out a comparative analysis of public sanitation policies in the European Union countries, meanly in Spain. We have set several recommendations and conclusions that could improve the Brazilian experience in the new federal Law's renewed context for basic sanitation.

**Author Contributions:** Both authors made substantial contributions to the conception and design of the work; and the acquisition, analysis, or interpretation of data. Both authors agree to be personally accountable for the author's own contributions and for ensuring that questions related to the accuracy or integrity of any part of the work, even ones in which the author was not personally involved, are appropriately investigated, resolved, and documented in the literature. Conceptualization—A.N.O.; Methodology—A.N.O., R.B.N.; Formal Analysis—A.N.O., R.B.N.; Data Curation—A.N.O., R.B.N.; Investigation—A.N.O., R.B.N.; Resources—A.N.O., R.B.N.; Writing–Original draft preparation—A.N.O., R.B.N.; Writing—Review & Editing, A.N.O.; Supervision, A.N.O.; Project Administration—A.N.O.; Funding Acquisition—A.N.O., R.B.N. All authors have read and agreed to the published version of the manuscript.

**Funding:** This research was supported by the CampusHabitat5U network of excellence of Valencian public universities (https://iuaca.ua.es/es/campushabitat5u.html) and the Cátedra del Agua of the University of Alicante, the Diputación Provincial de Alicante (https://catedradelaguaua.org/). This paper was carried out under the research projects (1) "Urban water services in Spain and Europe: return to direct management or public-private collaboration but with an effective government and public control of the service" (DER2017-87789-R) financed by the Spanish Ministry of Science, Innovation and Universities, and (2) "The Urban Water Cycle in Situations of Drought and Scarcity: Preventive and Reactive Measures" (B-SEJ-444-UGR18) financed by the Ministry of Economy and Knowledge of the Junta de Andalucía and the FEDER Andalusia Operational Program 2014-2020. Both projects are led by Estanislao Arana García, from the University of Granada.

**Institutional Review Board Statement:** Not applicable.

**Informed Consent Statement:** Not applicable.

**Data Availability Statement:** The data presented in this study is available on request from the corresponding author.

**Acknowledgments:** We thank Joaquín Melgarejo Moreno for the support.

**Conflicts of Interest:** The authors declare no conflict of interest.

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
