# Peer review of "Legal Aspects of Urban Water and Sanitation Regulatory Services: An Analysis of How the Spanish Experience Positively Would Contribute to the Brazilian New Regulation"

_water, doi:10.3390/w13081023_

Round 1

Reviewer 1 Report

Comments included in the attatched file.

Author Response

Comments included in the attatched file.

Reviewer 2 Report

The authors cover a relevant topic (WASH in urban contexts) in a resourceful manner. The paper assembles a lot of information on both cases.

Yet, the analytical and methodological focus of the paper remains unclear. What is the specific and novel contribution the authors want to make to which scientific debate? Since their analytical focus is unclear, it is hard to see what criteria they use to arrive at their results.

The paper also needs a language check.

My advice is to reject the paper and ask the authors to make their contribution to the scientific debate explicit and resubmit a revised paper.

Author Response

(The authors gave the same response as above.)

Reviewer 3 Report

I have read with interest your paper on Urban water governance in Brazil and Spain: a comparative analysis of water sanitation service. This is an important topic that deserve research. Here are some comments for improving it:

  • In the introduction clearly state: 1) the relevant literature within which this paper is situated; 2) the gap and your original contribution to the literature.
  • Explain again in the Methodological section why you have picked these two cases; justify them in a more convincing way.
  • Concerning Brazil, include the issue of transboundary waters and how these are incorporated in teh national directives (think of the different scales issue, and how waters are subject to different governance structures and dynamics in the case of shared groundwater - which are more locally used and decided upon - and surface waters; e.g. Guarani VS La Plata). I suggest reading and including the following two studies: 1)  da Silva, Luis Paulo Batista. "Production of scale in regional hydropolitics: an analysis of La Plata River Basin and the Guarani Aquifer System in South America." Geoforum 99 (2019): 42-53; 2) Hussein, H. (2018). The Guarani Aquifer System, highly present but not high profile: A hydropolitical analysis of transboundary groundwater governance. Environmental Science & Policy83, 54-62.
  • Discussion section: try to bring up the comparative element of the two case studies a bit more holistically, and show us similarities and differences

I hope these comments help in revising and strengthening this nice paper.

Author Response

(The authors gave the same response as above.)

Round 2

Reviewer 1 Report

I enclose my second revision. 

Author Response

Thank you very much for your second revision. 

We enclose our second revision to your comments.

Reviewer 2 Report

The authors made considerable changes to their paper. Yet, I am still not convinced by the research design. 

The text still does not sufficiently justify why Brazil, with a different political anf legal culture, size and population, and very different water resources etc., can learn from the Spanish case. The authors mention that they are aware of the complexity and difficulties of such a comparison; however, they do not explain how they will methodically cope with these challenges (page 4, lines 195f.). Without such a justification, I find the comparison and thus the results methodologically questionable. I can compare everything with everything.

Author Response

Thank you for your second revision. 

We enclose our revision to your comments.

Reviewer 3 Report

I would like to congratulate with the authors for the improvement in this paper. It now reads nicely, and much more improved.

In particular, the paper now engages with the relevant literature on the topic, both on the field/theme and theoretical, as well as with the regional and area studies literature on the topic. 

The methodology provides a clear picture of the methods used and a good justification.

Moreover, the originality of the paper is well explained in the introduction and the discussion brings together the intro, literature, and empirics. 

I am happy to say I have no further comments or critiques to suggest. I am also sure this paper will make a good contribution to the literature and to the broader debates in academia. 

Author Response

Thank you very much for your second revision, positive feedback and good vibes. Best regards.